# Effects of Hypoxia and Reoxygenation on Hypoxia-Responsive Genes, Physiological and Biochemical Indices in Hybrid Catfish (*Pelteobagrus vachelli* ♀ × *Leiocassis longirostris* ♂)

**DOI:** 10.3390/biology14080915

**Published:** 2025-07-23

**Authors:** Jie Yan, Faling Zhang, Fenfei Liang, Cheng Zhao, Shaowu Yin, Guosong Zhang

**Affiliations:** 1College of Marine Science and Engineering, Nanjing Normal University, Nanjing 210098, China; yanjie2547@163.com (J.Y.); 18383465626@163.com (F.Z.); zhaochengfish@foxmail.com (C.Z.); 2School of Agriculture and Bioengineering, Heze University, Heze 274015, China; fenfeiliang@163.com; 3Co-Innovation Center for Marine Bio-Industry Technology, Lian Yungang 222005, China

**Keywords:** hybrid catfish, hypoxic stress, reoxygenation, metabolic reprogramming, oxidative stress, mitochondrial function, apoptosis

## Abstract

This study explored how hybrid catfish respond to low oxygen levels (hypoxia) and subsequent recovery (reoxygenation), a common challenge in aquatic environments due to factors like pollution and climate change. We examined changes in genes and physiological processes in the brain, liver, and muscle tissues of the fish under controlled hypoxia and reoxygenation conditions. We found that low oxygen triggered adaptive responses, such as switching energy production from aerobic to anaerobic pathways in the liver, while the brain reduced anaerobic activity to avoid harmful lactate buildup. Muscle tissue showed the most severe effects, including oxidative stress, mitochondrial damage, and increased cell death. After oxygen levels were restored, the fish recovered, but signs of stress lingered in some tissues. These findings help explain how hybrid catfish tolerate fluctuating oxygen levels, which is crucial for improving aquaculture practices and ensuring fish health in changing environments. The study provides insights that could aid in breeding more resilient fish species, supporting sustainable fisheries and food security.

## 1. Introduction

Dissolved oxygen (DO) content in aquatic ecosystems is a critical factor for the survival, growth, and development of aquatic organisms [1,2]. However, DO concentration in aquatic ecosystems exhibits greater spatial and temporal variability compared to terrestrial environments due to factors such as global warming, eutrophication, pollution, photosynthesis, temperature, salinity, and seasonal variations [3]. Fish, as aquatic organisms, are highly susceptible to low-oxygen stress, and severe hypoxia can lead to large-scale mortality, resulting in a decline in fishery production. Therefore, hypoxia in aquatic ecosystems has become a significant concern in aquaculture and fishery science research.

Hypoxic conditions can significantly impact various physiological processes in fish, including behavior, growth, development, metabolism, and immune responses, potentially leading to mortality. To cope with dynamic fluctuations in dissolved oxygen (DO) concentrations, fish have evolved sophisticated adaptive mechanisms through long-term evolutionary processes [4,5]. For instance, fish can enhance hypoxia tolerance by regulating the transcription of relevant genes through the hypoxia-inducible factor (HIF) signaling pathway [6]. As a core component of oxygen-sensing proteins, HIF-1α is activated under hypoxic conditions and subsequently modulates the expression of downstream target genes, such as glycolysis-related genes (phosphofructokinase, liver type (*PFKL*), hexokinase 1 (*HK1*), pyruvate kinase (*PK*), lactate dehydrogenase A (*LDHA*)) and the tricarboxylic acid (TCA) cycle key enzyme gene (citrate synthase (CS)), thereby adjusting energy metabolism patterns [7,8]. Studies have demonstrated significant upregulation of HIF-1αA gene expression in the white muscle tissue of *Gymnocypris eckloni* under hypoxic conditions [9]. Similarly, *Hexagrammos otakii* exhibited markedly elevated serum HIF-1α levels following hypoxic stress [10]. Furthermore, fish can shift their respiratory metabolic mode from aerobic to anaerobic respiration to sustain energy supply under oxygen deprivation. For example, *Oncorhynchus mykiss* [11] and *Megalobrama amblycephala* [12] displayed reduced aerobic respiration rates, increased lactate concentrations, and decreased adenosine triphosphate (ATP) production rates under hypoxic conditions. Concurrently, fish enhance their antioxidant capacity to mitigate oxidative damage. Key enzymes in the antioxidant defense system, including superoxide dismutase (SOD), catalase (CAT), and glutathione peroxidase (GSH-PX), scavenge reactive oxygen species (ROS) and minimize oxidative injury [13,14]. Notably, *Larimichthys crocea* [15] showed significantly elevated activities of SOD, CAT, and GSH-PX under hypoxia, with similar antioxidant enzyme activity enhancements observed in *G. przewalskii* [16] and *Micropterus salmoides* [17]. These findings collectively demonstrate that fish achieve adaptive regulation to variable oxygen environments by activating a series of complex biological processes, thereby maintaining homeostasis.

Nevertheless, research on the adaptive mechanisms of hybrid fish to hypoxic environments, particularly the molecular regulatory mechanisms, remains relatively scarce. Significant variations exist in the hypoxic adaptability among different fish species, which may be attributed to their genetic backgrounds and physiological characteristics. For instance, darkbarbel catfish (*Pelteobagrus vachelli*) and longsnout catfish (*Leiocassis longirostris*), two species with distinct ecological adaptations, may produce hybrid offspring exhibiting enhanced hypoxia tolerance. *P. vachelli* belongs to the order Siluriformes, family Bagridae, and genus *Pelteobagrus*, while *L. longirostris* is classified under the same order and family but within the genus *Leiocassis*. The latter exhibits significantly faster growth rates and larger adult body sizes compared to the former. Interspecific hybridization breeding was conducted using 3-year-old *L. longirostris* as the paternal parent and *P. vachelli* as the maternal parent. The resulting hybrid offspring exhibited heterosis, demonstrating significantly faster growth rates compared to both parental species [18]. This study employs hybrid catfish as the experimental model to systematically investigate the effects of hypoxia stress and reoxygenation on oxygen-sensing proteins, respiratory metabolism, oxidative stress, apoptosis, inflammatory responses, and mitochondrial function in the brain, liver, and muscle tissues. The research aims to elucidate the dynamic changes in hypoxia-responsive genes and physiological–biochemical indicators, as well as to preliminarily clarify the molecular regulatory mechanisms underlying the responses of brain, liver, and muscle tissues to hypoxic stress.

## 2. Materials and Methods

### 2.1. Experimental Materials

The experimental 1-year-old hybrid yellow catfish (*Pelteobagrus vachelli* ♀ × *Leiocassis longirostris* ♂) were obtained from the Lukou Base of Nanjing Institute of Fisheries Sciences in Jiangsu Province, China (Nanjing, China). The fish had an average body weight of 20 ± 1.1 g and an average body length of 12 ± 0.7 cm. A total of 150 healthy and vigorous individuals were randomly allocated into three recirculating aquaculture glass tanks (equipped with biofiltration systems; dimensions: 1.2 m × 0.85 m × 0.55 m), with 50 fish per tank. The fish were acclimatized for 3 days under controlled conditions: water temperature was maintained at 25 ± 1 °C, dissolved oxygen (DO) at 7.0 mg/L, and pH at 7.5. During acclimation, the fish were fed twice daily (08:00 and 17:00) with formulated feed (Tongwei Yellow Catfish Floating Feed, Chengdu, China). This study was approved by the Nanjing Normal University Animal Ethics Committee (approval no. SYXK [Jiangsu] 2015-0028).

### 2.2. Hypoxic Stress Treatment

Prior to the formal experiments, preliminary trials were conducted to establish the hypoxia tolerance thresholds of *P*. *vachelli*, *L*. *longirostris*, and their hybrid under controlled temperature conditions (25 ± 1 °C). During the procedure, the aeration and water circulation systems in the aquaculture tanks were temporarily suspended, while high-purity nitrogen (99.99%) was introduced via a precision gas flow controller (Aosong Electronics, Guangzhou, China, AS200; regulation accuracy: ±0.05 L/min). The initial nitrogen flow rate was set at 3.0 L/min to rapidly reduce dissolved oxygen (DO) levels below 1.0 mg/L. As DO approached the target concentration (±0.2 mg/L), the flow rate was reduced to 0.8 L/min to prevent abrupt hypoxia-induced stress in the fish. Real-time DO monitoring was performed using an LDO101 (HACH, Singapore, Singapore, HQ1130) probe, with automated data logging at 10 s intervals. A PID feedback control system dynamically adjusted the nitrogen–air mixture ratio to maintain DO fluctuations within ±0.03 mg/L. Threshold breaches triggered automatic compensatory measures: fine-tuning nitrogen flow (±0.1 L/min) or brief air infusion (0.5 L/min). Results demonstrated species-specific surfacing responses: *L. longirostris* surfaced at 0.64 mg/L, *P. vachelli* at 0.61 mg/L, and the hybrid at 0.60 mg/L. Each experiment was replicated three times to ensure data robustness.

Preliminary experiments determined the surfacing point of hybrid yellow catfish to be 0.60 mg/L DO. Accordingly, the hypoxic stress condition was set at 0.7 mg/L DO, ensuring sustained and stable hypoxic exposure. Experimental groups included a normoxic control (C, 7.0 mg/L DO); hypoxic groups (H2, H4, H6 at 0.7 mg/L DO for 2, 4, and 6 h, respectively); and reoxygenation groups (R2, R4, R6 at 7.0 mg/L DO for 2, 4, and 6 h post-hypoxia). Each group contained three biological replicates (*n* = 5 fish per replicate). Prior to hypoxia induction, 15 fish (5 from each of 3 tanks at 7.0 mg/L DO) were sampled as controls. Brain, liver, and muscle tissues were collected, rinsed with ice-cold physiological saline (0.8%), and pooled as composite samples (5 fish/sample) in 1.5 mL cryotubes before flash-freezing in liquid nitrogen and storage at −80 °C. Following the termination of aeration and water circulation systems, pure nitrogen gas was immediately infused into the recirculating aquaculture glass tanks to rapidly reduce dissolved oxygen (DO) to 0.7 mg/L within 30 min. The hypoxic exposure period commenced upon reaching the target DO level, which was continuously monitored using an oxygen meter. Precise regulation of nitrogen and air infusion rates maintained the desired DO concentration throughout the experiment. Biological sampling was performed at three hypoxic timepoints (2 h, 4 h, and 6 h), with 15 specimens collected at each interval using identical procedures to the control group. After 6 h of hypoxic exposure, nitrogen infusion was discontinued and the system was restored through activation of water circulation with atmospheric aeration, maintaining DO at 7.0 mg/L during reoxygenation. Subsequent sampling was conducted at 2 h, 4 h, and 6 h reoxygenation intervals (15 specimens per timepoint), adhering to the standardized sampling protocol.

### 2.3. Temporal Gene Expression Analysis

All brain, liver, and muscle tissue samples were retrieved from −80°C storage and homogenized into powder using liquid nitrogen in pre-chilled mortars. Total RNA was extracted from brain, liver, and muscle tissues using the FastPure RNA Kit (Nanjing Vazyme, China RC101-01, Nanjing, China). Briefly, 10–20 mg of tissue was homogenized in 500 μL Buffer RL, followed by gDNA removal and RNA purification through column filtration. RNA quality was verified by spectrophotometry (OD260/280 ratio 1.8–2.0) and agarose gel electrophoresis. cDNA was synthesized with 1 μg total RNA using the Hifair^®^ III cDNA Synthesis Kit (Hubei Yeasen, Wuhan, China), including gDNA digestion (42 °C, 2 min) and reverse transcription (25 °C for 5 min, 55 °C for 15 min, 85 °C for 5 min). Gene-specific primers (Table 1) were designed with Premier 5.0, and qRT-PCR was performed using SYBR Green Master Mix (Hubei Yeasen, China Yeasen) under standard cycling conditions (95 °C for 5 min; 40 cycles of 95 °C/10 s and 60 °C/30 s). Relative expression was calculated via the 2^−ΔΔCt^ method with β-actin as the reference.

### 2.4. Enzyme Activity Assay

A precise weight of 0.1 g of tissue was measured and homogenized mechanically in a 1:9 weight-to-volume ratio with 0.9% physiological saline under ice-bath conditions to prepare a 10% homogenate. The homogenate was centrifuged at 4 °C and 2500 r/min for 10 min, and the supernatant was collected for enzyme activity determination. The enzyme activity assay kits used in this experiment, including those for hexokinase (HK), citrate synthase (CS), lactate dehydrogenase (LDH), pyruvate kinase (PK), lipid peroxidase (LPO), malondialdehyde (MDA), protein carbonyl (PCO), total superoxide dismutase (T-SOD), glutathione peroxidase (GSH-PX), and catalase (CAT), were all sourced from Nanjing Jiancheng Bioengineering Institute (Nanjing, China). The specific detection methods followed the instructions provided with the kits (Appendix A).

### 2.5. Data Processing

Statistical analysis of the data was performed using SPSS 27.0 software. One-way ANOVA and Dunnett *t*-tests were conducted to analyze significant differences between the control group and the hypoxia and reoxygenation groups. A *p*-value of less than 0.05 was considered statistically significant (denoted by *). Experimental data are presented as mean ± SD. Graphs were generated using GraphPad Prism 9 based on the analysis results.

## 3. Results

### 3.1. Effects of Hypoxia and Reoxygenation on Oxygen-Sensing Proteins and Respiratory Metabolism-Related Genes in Hybrid Catfish

Under acute hypoxia, the expression of the *HIF-1α* gene in brain tissue exhibited a significant upward trend, peaking at H6 (*p* < 0.05), with a 2.27-fold increase compared to the control group. This elevated expression remained significantly different from the control group at R2 (*p* < 0.05) before gradually returning to baseline levels. The *HK1* gene exhibited a fluctuating expression pattern, peaking at H2 (*p* < 0.05) with a 1.63-fold increase relative to the control group, followed by a decline. Upon reoxygenation, its expression rebounded and remained significantly elevated at R4 (*p* < 0.05) compared to the control. In contrast, the *PFKL*, *PK*, and *CS* genes showed no significant changes under hypoxic conditions (*p* > 0.05). However, the *PFKL* and *PK* genes were significantly upregulated at R2 and R4, respectively (*p* < 0.05), before returning to control levels. Conversely, the *LDHA* gene displayed a downward trend, with significant downregulation observed at H6 (*p* < 0.05), reaching 0.63 times the control level, and gradually returning to baseline levels after reoxygenation (Figure 1).

In liver tissue under acute hypoxia, the expression of the *HIF-1α* and *PFKL* genes initially increased, peaking at H4 and H2, respectively (*p* < 0.05), with 1.97- and 1.39-fold increases compared to the control group. These levels remained significantly elevated at R4 and R2, respectively (*p* < 0.05), following reoxygenation. The *HK1* and *PK* genes exhibited a fluctuating upward trend, while the *LDHA* gene showed a consistent upward trend, all reaching their peaks at H6 (*p* < 0.05), with 2.74-, 3.31-, and 3.03-fold increases relative to the control group, respectively. These levels remained significantly higher than the control group at R2 (*p* < 0.05) after reoxygenation. In contrast, the *CS* gene showed no significant changes under either hypoxic or reoxygenation conditions (Figure 2).

In muscle tissue under acute hypoxia, the expression of the *HIF-1α* and *HK1* genes initially increased, peaking at H4, and then decreased to a minimum at H6 (*p* < 0.05), with 2.58- and 0.50-fold changes relative to the control group, respectively. These levels remained significantly elevated at R2 and R4 (*p* < 0.05) following reoxygenation. The *PFKL* gene exhibited an initial increase followed by a decrease, showing no significant changes compared to the control group. However, it was significantly upregulated at R2 after reoxygenation before returning to baseline levels. Similarly, the *PK* gene initially increased and then decreased, showing no significant changes compared to the control group, but was significantly higher than the control group at R2 (*p* < 0.05) after reoxygenation before returning to baseline levels. In contrast, the *CS* and *LDHA* genes both displayed a downward trend, with significant downregulation observed at H6 (*p* < 0.05), reaching 0.57 and 0.72 times the control levels, respectively. After reoxygenation, the *CS* gene was significantly upregulated compared to the control group (*p* < 0.05), while the *LDHA* gene showed no significant changes (Figure 3).

### 3.2. Effects of Hypoxia and Reoxygenation on Respiratory Metabolism-Related Enzyme Activities in Hybrid Catfish

Under acute hypoxia, the activity of pyruvate kinase (PK) in brain tissue initially increased, peaking at H2 (*p* < 0.05), reaching 1.45 times the level of the control group, followed by a gradual decline and eventual return to baseline levels after reoxygenation. In contrast, the activities of hexokinase (HK) and lactate dehydrogenase (LDH) exhibited an initial decrease followed by an increase. Although HK activity did not significantly differ from the control group under hypoxic conditions, it peaked at R6 following reoxygenation, demonstrating a significant difference compared to the control group (*p* < 0.05). LDH activity reached its minimum at H6, decreasing to 0.56 times the control level, but no significant difference was observed compared to the control group after reoxygenation. Citrate synthase (CS) activity displayed a downward trend during hypoxia and an upward trend following reoxygenation, with no significant differences observed compared to the control group under either condition (Figure 4).

In liver tissue under acute hypoxia, the activities of pyruvate kinase (PK) and lactate dehydrogenase (LDH) generally exhibited an upward trend, peaking at H6 (*p* < 0.05), reaching 4.16 and 1.53 times the levels of the control group (C), respectively. Following reoxygenation, PK activity peaked at R4, demonstrating a significant difference compared to the control group (*p* < 0.05), whereas LDH activity showed no significant difference. Hexokinase (HK) activity initially increased, significantly upregulated at H2 (*p* < 0.05), and peaked at H4, reaching 2.76 times the control level (C), before gradually returning to baseline levels after reoxygenation. The citrate synthase (CS) activity exhibited an initial decline followed by a subsequent increase, reaching its minimum value at H4 (*p* < 0.05), which was 0.71-fold of the control group (C). The activity gradually recovered to control levels following reoxygenation (Figure 5).

In muscle tissue under acute hypoxia, hexokinase (HK) activity initially increased, reaching its minimum at H6 (*p* < 0.05), decreasing to 0.56 times the control level (C), and gradually returned to baseline levels after reoxygenation. Citrate synthase (CS) activity exhibited a downward trend, reaching its minimum at H6, decreasing to 0.60 times the control level (C), and subsequently peaked at R2 after reoxygenation, demonstrating a significant difference compared to the control group (*p* < 0.05). The activities of pyruvate kinase (PK) and lactate dehydrogenase (LDH) exhibited no significant differences compared to the control group under either hypoxic or reoxygenation conditions (Figure 6).

### 3.3. Effects of Hypoxia and Reoxygenation on Oxidative Stress-Related Genes in Hybrid Catfish

Under acute hypoxia, the expression levels of *SOD1* and *GSH-PX* genes in brain tissue exhibited a biphasic response, showing an initial decrease followed by an increase, with minimum levels observed at H4 (*p* < 0.05), corresponding to 0.51- and 0.69-fold changes relative to the control group (C), respectively. Following reoxygenation, both genes demonstrated peak expression at R4, with statistically significant differences compared to the control group (*p* < 0.05). The expression pattern of the *SOD2* gene displayed a progressive increase, whereas *CAT* gene expression demonstrated a decreasing trend. Both genes attained their maximum expression levels at R2 post-reoxygenation, with values significantly elevated compared to the control group (*p* < 0.05) (Figure 7).

In liver tissue under acute hypoxia, the expression levels of *SOD1* and *CAT* genes showed an increasing trend, reaching peak levels at H6 (*p* < 0.05), with 1.79- and 2.05-fold increases relative to the control group (C), respectively. During the reoxygenation phase, *SOD1* expression remained significantly elevated compared to control levels, while *CAT* expression peaked at R2, showing a significant increase over control values (*p* < 0.05), before gradually returning to baseline levels. The expression profiles of *SOD2* and *GSH-PX* genes exhibited an initial increase followed by a decrease, with maximum levels observed at H4 (*p* < 0.05), corresponding to 2.0- and 1.46-fold changes relative to control levels, respectively. Following reoxygenation, these genes demonstrated peak expression at R2 and R4, respectively, showing statistically significant differences from control values (*p* < 0.05), before gradually returning to baseline levels (Figure 8).

In muscle tissue subjected to acute hypoxia, the expression levels of *SOD1*, *SOD2*, and *CAT* genes displayed an initial increase followed by a decrease, reaching peak levels at H4 (*p* < 0.05), with 2.80-, 2.59-, and 2.32-fold increases relative to control values, respectively. During the reoxygenation phase, *SOD1* expression showed a gradual decline while maintaining significantly higher levels than the control group, whereas *SOD2* and *CAT* expression levels progressively increased, peaking at R6 and showing statistically significant differences from control values (*p* < 0.05). The *GSH-PX* gene expression profile demonstrated a continuous upward trend, reaching maximum levels at H6 (*p* < 0.05), representing a 2.70-fold increase over control values, and remained significantly elevated compared to control levels throughout the reoxygenation period (*p* < 0.05) (Figure 9).

### 3.4. Effects of Hypoxia and Reoxygenation on Oxidative Stress Enzyme Activities and Parameters in Hybrid Catfish

Under acute hypoxia, the activities of T-SOD and GSH-PX in brain tissue decreased significantly, reaching their lowest levels at H6 (*p* < 0.05), representing 0.66 and 0.54 times the levels of the control group (C), respectively. After reoxygenation, these activities remained significantly lower than the control group at R2, after which they gradually increased and returned to control levels. The activities of CAT and PCO declined with fluctuations, with only PCO activity significantly reduced at H6 (*p* < 0.05), reaching 0.55 times the control level (C). After reoxygenation, only CAT activity showed a gradual increase, peaking at R6 and demonstrating a significant difference from the control group (*p* < 0.05). The activities of LPO and MDA initially rose and then declined, with only MDA activity significantly reduced at H6 (*p* < 0.05), reaching 0.65 times the control level (C). After reoxygenation, only LPO activity reached its peak at R2, after which it gradually decreased and returned to control levels (Figure 10).

In liver tissue under acute hypoxia, the activities of T-SOD, CAT, LPO, and MDA increased significantly, peaking at H6 (*p* < 0.05), reaching 2.05, 1.41, 2.0, and 2.38 times the levels of the control group (C), respectively. After reoxygenation, the activities of T-SOD, CAT, LPO, and MDA reached their peaks at R4, R2, R4, and R6, respectively. While CAT activity gradually returned to control levels, the activities of the other enzymes remained significantly elevated compared to the control group (*p* < 0.05). The activities of GSH-PX and PC exhibited an increasing trend, both reaching their peak values at H6 (*p* < 0.05), which were 2.27-fold and 1.39-fold of the control group (C), respectively. After reoxygenation, GSH-PX activity showed a gradual increase, peaking at R6 and demonstrating a significant difference from the control group (*p* < 0.05), while PCO activity reached its peak at R2, significantly elevated compared to the control group (*p* < 0.05), before gradually declining and returning to control levels (Figure 11).

In muscle tissue under acute hypoxia, the activities of T-SOD, CAT, LPO, MDA, and PCO increased significantly, peaking at H6 (*p* < 0.05), reaching 2.41, 1.32, 2.03, 3.74, and 3.81 times the levels of the control group (C), respectively. After reoxygenation, the activities of T-SOD, CAT, and MDA reached their peaks at R2, demonstrating significant differences from the control group (*p* < 0.05), before gradually declining and returning to control levels. In contrast, the activities of LPO and PCO remained significantly elevated compared to the control group after reoxygenation (*p* < 0.05). The activity of glutathione peroxidase (GSH-PX) demonstrated a significant upward trend, peaking at H6 (*p* < 0.05) with levels reaching 2.52-fold of the control group (C). Notably, during the reoxygenation phase, GSH-PX activity remained significantly elevated compared to control values (*p* < 0.05) (Figure 12).

### 3.5. Effects of Hypoxia and Reoxygenation on Mitochondrial Damage Genes in Hybrid Catfish

During acute hypoxia and subsequent reoxygenation, the expression levels of *PGC-1α*, *COXIV*, and *ATP5A1* genes in brain tissue remained unchanged compared to the control group (Figure 13).

Under acute hypoxia, the expression of *PGC-1α* and *COXIV* genes in liver tissue decreased significantly, reaching their minimum values at H6 (*p* < 0.05), which were 0.59 and 0.58 times that of the control group, respectively. After reoxygenation, the expression of *PGC-1α* and *COXIV* genes recovered gradually to control levels, although *COXIV* expression remained significantly lower than the control group at R2 (*p* < 0.05). The *ATP5A1* gene expression initially decreased, reaching its minimum value at H4 (*p* < 0.05), which was 0.58 times that of the control group, and subsequently increased to control levels after reoxygenation (Figure 14).

Under acute hypoxia, the expression of the *PGC-1α* gene in muscle tissue decreased significantly, reaching its minimum value at H6 (*p* < 0.05), which was 0.42 times that of the control group. After reoxygenation, *PGC-1α* expression remained significantly lower than the control group at R2 (*p* < 0.05) but gradually recovered to control levels thereafter. The expression of *COXIV* and *ATP5A1* genes initially decreased, reaching their minimum values at H4 (*p* < 0.05), which were 0.27 and 0.38 times that of the control group (C), respectively. After reoxygenation, their expression remained significantly lower than the control group at R2 (*p* < 0.05) but gradually recovered to control levels (Figure 15).

### 3.6. Effects of Hypoxia and Reoxygenation on Inflammation and Apoptosis-Related Genes in Hybrid Catfish

During acute hypoxia and subsequent reoxygenation, no significant differences were observed in the expression levels of *IL-1β*, *IKKβ*, *Caspase 3*, *Bax*, and *Bcl-2* genes in brain tissue when compared to the control group (Figure 16).

Under acute hypoxia, the expression levels of *IL-1β*, *IKKβ*, *Caspase 3*, and *Bax* genes in liver tissue significantly increased, reaching their maximum values at H6 (*p* < 0.05), with expression levels 1.95-, 1.96-, 2.00-, and 2.13-fold higher than those of the control group (C), respectively. After reoxygenation, the expression levels of these genes gradually decreased, returning to control levels, although *IL-1β*, *IKKβ*, and Bax remained significantly elevated compared to the control group at R2 (*p* < 0.05). In contrast, the expression of the *Bcl-2* gene significantly decreased, reaching its minimum value at H6 (*p* < 0.05), with expression levels 0.66-fold lower than those of the control group (C), and subsequently increased to control levels following reoxygenation (Figure 17).

Under acute hypoxia, the expression of *IL-1β* and *IKKβ* genes in muscle tissue initially increased, peaking at H4 (*p* < 0.05), with expression levels 3.26- and 2.84-fold higher than those of the control group (C), respectively, before subsequently decreasing. After reoxygenation, the expression levels of these genes gradually declined, returning to control levels, although *IL-1β* remained significantly elevated compared to the control group at R2 (*p* < 0.05). The expression of the *Caspase 3* gene significantly increased, reaching its maximum value at H6 (*p* < 0.05), with expression levels 2.44-fold higher than those of the control group (C), and subsequently decreased to control levels following reoxygenation. The expression of the *Bax* gene fluctuated but overall increased, peaking at H6 (*p* < 0.05), with expression levels 2.30-fold higher than those of the control group (C), and remained significantly elevated compared to the control group at R2 (*p* < 0.05) before gradually declining to control levels. The expression of the *Bcl-2* gene initially decreased, reaching its minimum value at H4 (*p* < 0.05), with expression levels 0.54-fold lower than those of the control group (C), and subsequently increased to control levels following reoxygenation (Figure 18).

## 4. Discussion

### 4.1. Effects of Hypoxia and Reoxygenation on Oxygen-Sensing Protein HIF-1α and Respiratory Metabolism in Hybrid Catfish

The survival of aquatic organisms is primarily dependent on the stability of their aquatic environment, with fluctuations in dissolved oxygen (DO) levels being of particular significance for fish [19]. To cope with significant changes in DO, fish have evolved a series of complex physiological adaptation mechanisms, among which the activation of the *HIF-1* signaling pathway is of paramount importance [20]. This pathway, mediated by the binding of HIF heterodimers to hypoxia response elements (*HREs*), activates a cascade of genes involved in angiogenesis, glucose and iron transport, glycolysis, and cell cycle regulation, thereby enabling fish to maintain viability under hypoxic conditions [21]. The results of this study indicate that the expression of the *HIF-1α* gene in the brain, liver, and muscle tissues of hybrid catfish was significantly upregulated during hypoxia and subsequently returned to control levels after reoxygenation. These findings are consistent with studies on other fish species, such as *Sebastes schlegelii* [22], *Leiostomus xanthurus* [23], and *Oreochromis niloticus* [24], thereby reinforcing *HIF-1α* as a key oxygen-sensing protein that is significantly activated under hypoxic conditions.

Research demonstrates that the *HIF-1* signaling pathway regulates key genes such as *PFKL, HK1, PK, LDHA*, and *CS*, which are integral to glycolysis, anaerobic respiration, and the tricarboxylic acid (TCA) cycle [25,26,27,28]. Under hypoxia, organisms typically adapt by lowering basal metabolic rates and transitioning between aerobic and anaerobic metabolic pathways [29]. Although this adaptation may result in physiological burdens such as lactate accumulation, it is critical for survival [11]. In this study, glycolysis-related (*PFKL*, HK1, PK) and anaerobic respiration-related (*LDHA*) genes and enzyme activities in the liver of hybrid catfish were significantly upregulated under hypoxia and subsequently returned to normal levels after reoxygenation. These results align with findings in *P. fulvidraco* [30], *P. vachelli* [31], and hybrid yellow catfish (*P. fulvidraco* ♀ × *P. vachelli* ♂) [32], indicating that liver tissue adapts to hypoxia by enhancing glycolysis and anaerobic respiration to ensure energy supply. Additionally, in the brain tissue, the *HK* gene and PK enzyme activity were significantly upregulated at H2, suggesting the activation of glycolysis during the early stages of hypoxia. Notably, *LDHA* was significantly downregulated at H6, potentially representing a protective mechanism to prevent lactate accumulation. In contrast, some glycolysis-, CS-, and anaerobic respiration-related genes and enzyme activities in muscle tissue exhibited a downward trend, consistent with findings in *P. vachelli* [33,34] and *Trachinotus blochii* [35]. This implies that muscle tissue may experience more severe hypoxia-induced metabolic disturbances compared to other tissues.

### 4.2. Effects of Hypoxia and Reoxygenation on Oxidative Stress in Hybrid Catfish

Under physiological homeostasis, the generation and clearance of reactive oxygen species (ROS) maintain a delicate dynamic equilibrium [36]. However, exposure to environmental stressors can disrupt this equilibrium, resulting in oxidative stress [37]. Numerous studies have demonstrated that various aquatic organisms, including *Carassius auratus* [38], *Sinonovacula constricta* [39], *Megalobrama amblycephala* [40], and *Neohelice granulata* [41], exhibit significant upregulation of antioxidant enzyme activities under hypoxic conditions, a phenomenon termed “oxidative stress preparation” to mitigate potential oxidative stress during reoxygenation [42]. Similarly, hybrid catfish exhibited analogous alterations in antioxidant enzyme activities under hypoxic conditions. Our findings revealed that antioxidant enzymes (e.g., SOD, CAT, GSH-PX) in hepatic and muscular tissues exhibited significant increases during the initial hypoxic phase (H2–H4), whereas oxidative stress markers (e.g., LPO, MDA, PCO) demonstrated substantial elevation during H4-H6. These observations suggest that hybrid catfish may possess an anticipatory antioxidant defense mechanism to counteract potential oxidative stress induced by hypoxic conditions [43]. In contrast, both antioxidant enzymes and oxidative stress markers in cerebral tissue remained stable or exhibited minimal declines, maintaining relatively consistent levels. Comparable findings have been documented in *P. glenii* [44] and *P. vachelli* [31], suggesting that cerebral tissue, as a vital organ, may receive preferential oxygen supply during acute hypoxic conditions, thereby minimizing oxidative stress damage [45,46]. Notably, oxidative stress markers in hepatic and muscular tissues demonstrated significant elevation during the advanced hypoxic phase, with muscular tissue displaying higher oxidative stress levels compared to hepatic tissue, indicating more pronounced oxidative damage in muscular tissue.

Reoxygenation has been demonstrated to induce rapid ROS accumulation, resulting in secondary oxidative stress responses. For instance, *Cyprinus carpio var qingtianensis* [47] and *Mugil cephalus* [48] demonstrated significant oxidative stress during reoxygenation. Similarly, the hybrid catfish exhibited ROS accumulation during reoxygenation. The present study revealed that antioxidant enzyme activities and oxidative stress parameters in brain, liver, and muscle tissues were significantly elevated following reoxygenation, suggesting that the antioxidant enzyme system maintains high activity to mitigate reoxygenation-induced oxidative stress.

### 4.3. Effects of Hypoxia and Reoxygenation on Mitochondrial Damage, Inflammation, and Apoptosis in Hybrid Catfish

Mitochondria, serving as the central organelles for cellular energy metabolism, predominantly depend on oxidative phosphorylation for energy conversion [49]. Numerous studies have demonstrated that hypoxia or ischemia significantly impairs mitochondrial oxidative phosphorylation, resulting in diminished mitochondrial biogenesis and functional impairments [50]. Key regulatory genes including *PGC-1α*, *COXIV*, and *ATP5A1* are intrinsically linked to mitochondrial function, orchestrating mitochondrial biogenesis and maintenance, while their expression profiles serve as reliable biomarkers for mitochondrial dysfunction [51,52,53,54]. These genes play pivotal roles in regulating oxidative phosphorylation, thereby directly modulating cellular energy metabolism and maintaining functional homeostasis [55]. Experimental studies conducted on rat models [56,57] and *P. vachelli* [34] have consistently demonstrated that hypoxia significantly downregulates the expression of *PGC-1α*, *COXIV*, and *ATP5A1* genes, highlighting the profound detrimental effects of hypoxia on mitochondrial functionality. In the current investigation, we observed significant downregulation of mitochondrial-related genes (*PGC-1α*, *COXIV*, *ATP5A1*) in the liver and muscle tissues of hybrid catfish under hypoxic conditions, suggesting substantial impairment in mitochondrial respiration and biogenesis processes. However, in brain tissue, these genes maintained stable expression levels, suggesting the presence of robust protective mechanisms and adaptive responses that preserve mitochondrial function under hypoxic stress.

The NF-κB signaling pathway plays a pivotal role in regulating inflammatory signaling transduction and mediating the expression of inflammatory factors [58]. This signaling cascade modulates the transcriptional activation of specific inflammatory factors, including TNF-α and IL-1β, in response to various cellular stress stimuli [59]. In *P. vachelli* [45], hypoxia-induced mitochondrial dysfunction was found to activate the NF-κB signaling pathway, subsequently triggering the release of pro-inflammatory cytokines and initiating apoptotic processes. Our experimental results demonstrated that the expression levels of *IKKβ*, *IL-1β*, *Caspase-3*, and *Bax* genes in muscle tissue were significantly upregulated during the initial phase of hypoxia exposure, showing more pronounced elevation compared to hepatic tissue, indicating that muscle cells might exhibit enhanced susceptibility to hypoxia-induced apoptosis. Furthermore, the elevated expression of *IKKβ*, a pivotal regulatory component of the NF-κB signaling cascade, was observed in both hepatic and muscular tissues, providing additional evidence for the pathway’s essential involvement in hypoxia-mediated cellular stress responses. Notably, the pronounced upregulation of *IKKβ* in muscular tissue suggests a potentially heightened cellular sensitivity to hypoxic conditions. Comparable observations have been documented in *Rachycentron canadum* [60], *L. crocea* [61], and *Epinephelus coioides* [62], where hypoxia-triggered reactive oxygen species (ROS) and peroxide accumulation were shown to activate the NF-κB signaling cascade [63], resulting in the upregulation of inflammatory mediators (*TNF-α*, *IL-1β*) and pro-apoptotic factors (*Caspase-3*, *Bax*), concomitant with downregulation of anti-apoptotic Bcl-2 expression. In the current investigation, we observed significant upregulation of *IL-1β*, *Caspase-3*, and *Bax* gene expression in both hepatic and muscular tissues under hypoxic conditions, accompanied by downregulation of *Bcl-2* expression, which is presumably attributable to excessive ROS generation and subsequent NF-κB pathway activation. Conversely, the expression profiles of these inflammatory and apoptotic markers remained relatively stable in cerebral tissue, in agreement with previous findings in *P. vachelli* [33], implying that neural tissue might possess specialized metabolic adaptations and enhanced antioxidant defense mechanisms to counteract hypoxia-mediated cellular injury.

## 5. Conclusions

This study systematically investigated the effects of hypoxia–reoxygenation on oxygen-sensing proteins, respiratory metabolism, oxidative stress, mitochondrial function, inflammatory responses, and apoptosis in the brain, liver, and muscle tissues of hybrid catfish. Hypoxia significantly activated HIF-1α expression and induced metabolic reprogramming, with the liver shifting toward anaerobic metabolism, while the brain exhibited suppressed glycolysis and the muscle displayed concurrent inhibition of glycolysis and aerobic respiration. Notably, muscle tissue demonstrated heightened oxidative stress, mitochondrial dysfunction, inflammatory cascades, and apoptosis during hypoxia–reoxygenation. These findings elucidate the multi-tissue adaptive mechanisms in hybrid catfish under oxygen fluctuations. Future research should explore interspecific variations in hypoxia adaptation and genetic strategies to enhance hypoxia tolerance, providing insights for sustainable aquaculture development.

## Figures and Tables

**Figure 1 biology-14-00915-f001:**
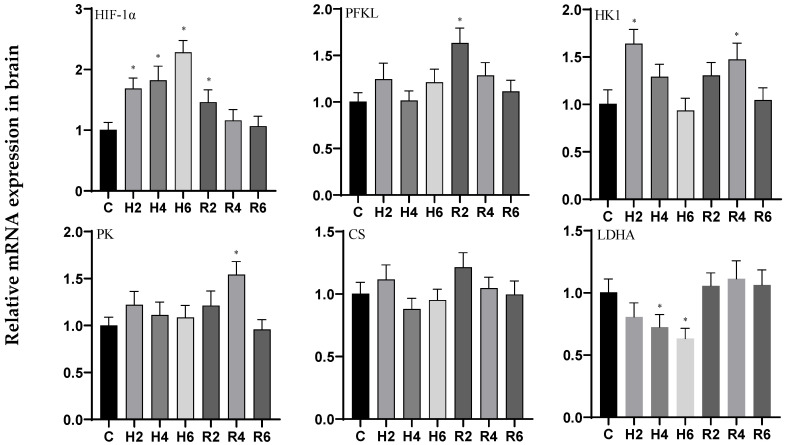
Effects of hypoxia (2 h, 4 h, and 6 h) and reoxygenation (2 h, 4 h, and 6 h) on brain oxygen-sensing protein and respiratory metabolism-related genes of hybrid catfish. Note: * at *p* < 0.05.

**Figure 2 biology-14-00915-f002:**
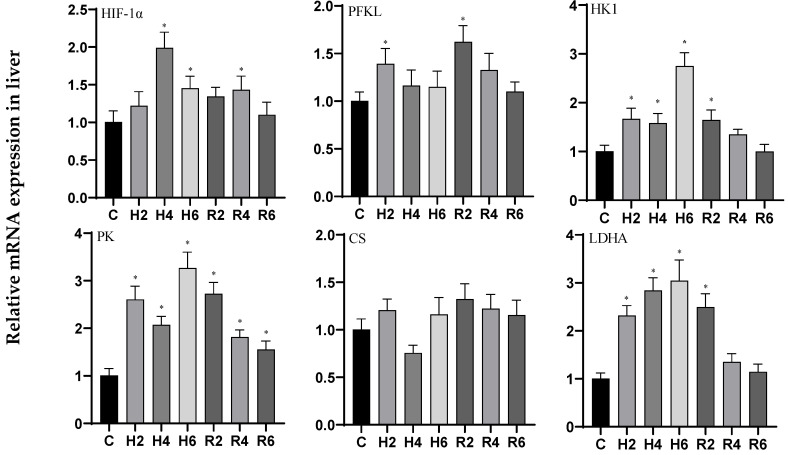
Effects of hypoxia (2 h, 4 h, 6 h) and reoxygenation (2 h, 4 h, 6 h) on liver oxygen-sensing protein and respiratory metabolism-related genes of hybrid catfish. Note: * at *p* < 0.05.

**Figure 3 biology-14-00915-f003:**
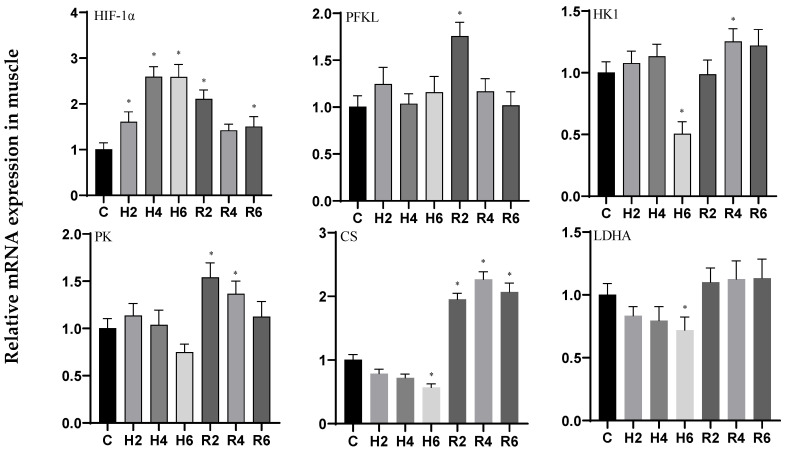
Effects of hypoxia (2 h, 4 h, 6 h) and reoxygenation (2 h, 4 h, 6 h) on muscle oxygen-sensing protein and respiratory metabolism-related genes of hybrid catfish. Note: * at *p* < 0.05.

**Figure 4 biology-14-00915-f004:**
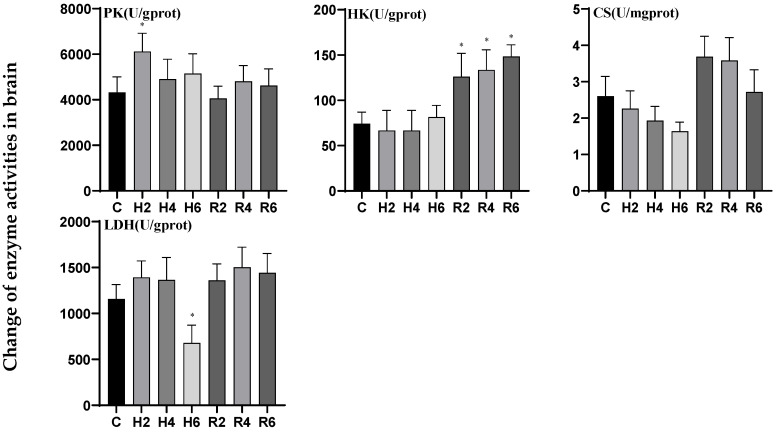
Effects of hypoxia (2 h, 4 h, 6 h) and reoxygenation (2 h, 4 h, 6 h) on brain respiratory metabolism-related enzyme activities of hybrid catfish. Note: * at *p* < 0.05.

**Figure 5 biology-14-00915-f005:**
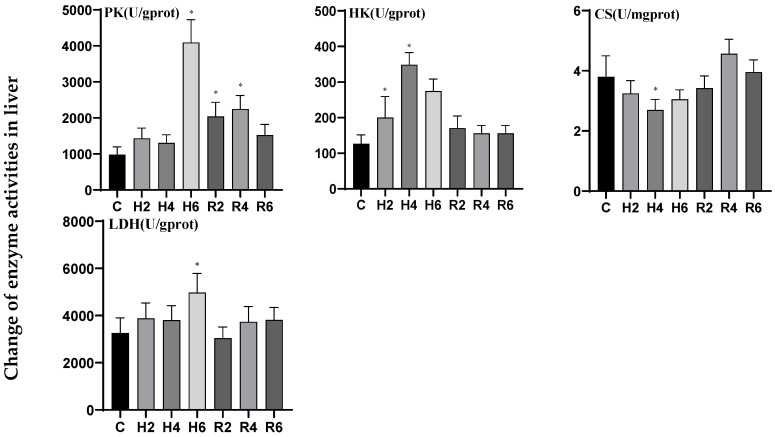
Effects of hypoxia (2 h, 4 h, 6 h) and reoxygenation (2 h, 4 h, 6 h) on liver respiratory metabolism-related enzyme activities of hybrid catfish. Note: * at *p* < 0.05.

**Figure 6 biology-14-00915-f006:**
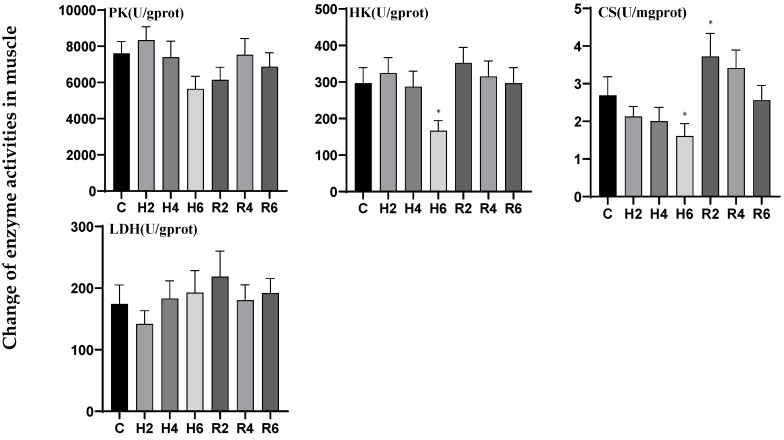
Effects of hypoxia (2 h, 4 h, 6 h) and reoxygenation (2 h, 4 h, 6 h) on muscle respiratory metabolism-related enzyme activities of hybrid catfish. Note: * at *p* < 0.05.

**Figure 7 biology-14-00915-f007:**
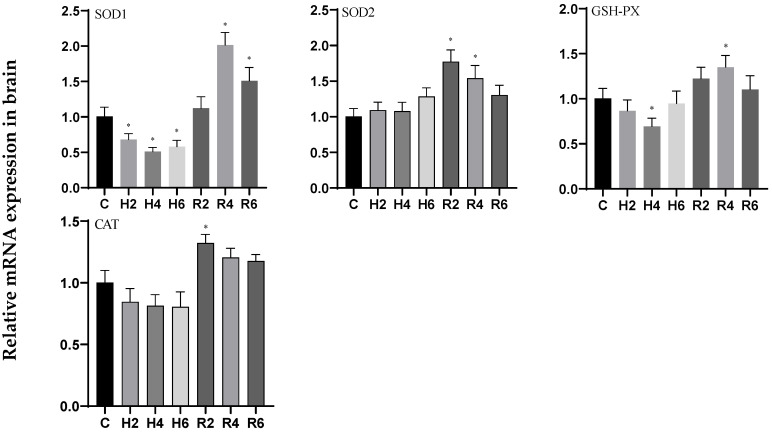
Effects of hypoxia (2 h, 4 h, 6 h) and reoxygenation (2 h, 4 h, 6 h) on oxidative stress-related genes in brain of hybrid catfish. Note: * at *p* < 0.05.

**Figure 8 biology-14-00915-f008:**
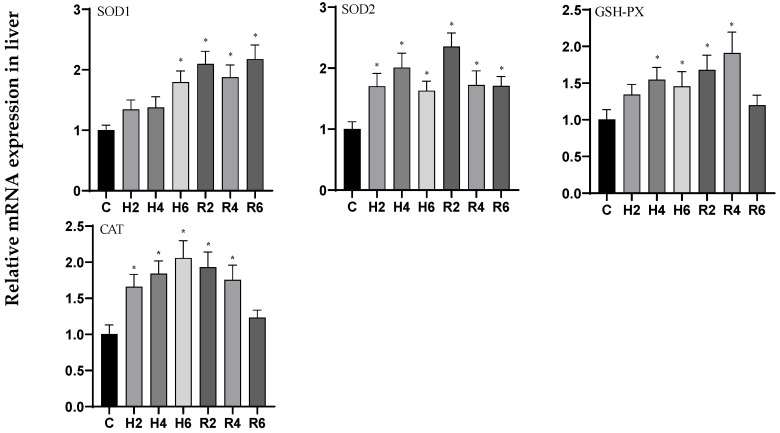
Effects of hypoxia (2 h, 4 h, 6 h) and reoxygenation (2 h, 4 h, 6 h) on oxidative stress-related genes in liver of hybrid catfish. Note: * at *p* < 0.05.

**Figure 9 biology-14-00915-f009:**
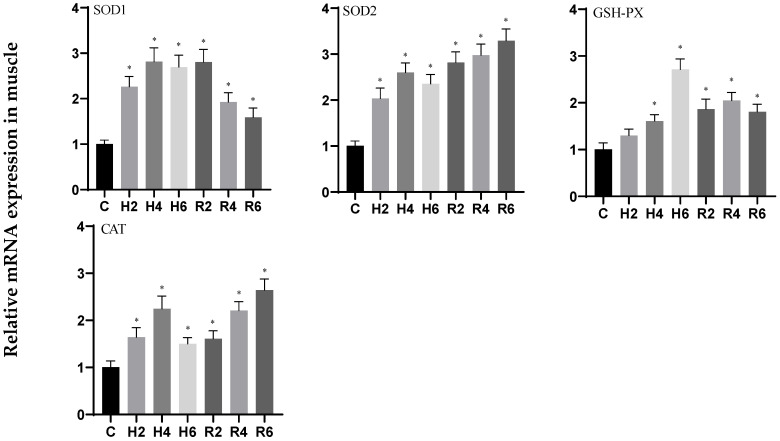
Effects of hypoxia (2 h, 4 h, 6 h) and reoxygenation (2 h, 4 h, 6 h) on oxidative stress-related genes in muscle of hybrid catfish. Note: * at *p* < 0.05.

**Figure 10 biology-14-00915-f010:**
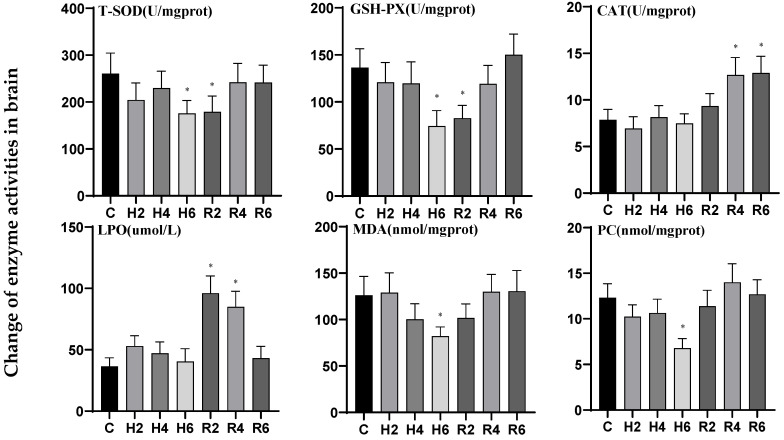
Effects of hypoxia (2 h, 4 h, 6 h) and reoxygenation (2 h, 4 h, 6 h) on brain oxidative kinase activity and parameters of hybrid catfish. Note: * at *p* < 0.05.

**Figure 11 biology-14-00915-f011:**
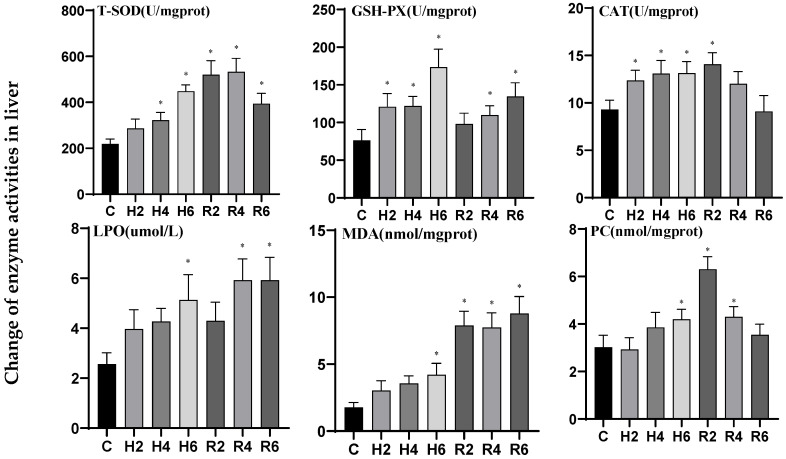
Effects of hypoxia (2 h, 4 h, 6 h) and reoxygenation (2 h, 4 h, 6 h) on liver oxidative kinase activity and parameters of hybrid catfish. Note: * at *p* < 0.05.

**Figure 12 biology-14-00915-f012:**
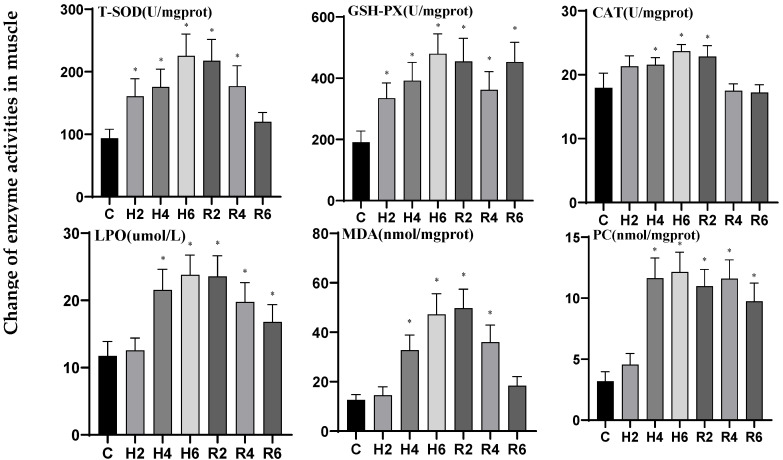
Effects of hypoxia (2 h, 4 h, 6 h) and reoxygenation (2 h, 4 h, 6 h) on muscle oxidative kinase activity and parameters of hybrid catfish. Note: * at *p* < 0.05.

**Figure 13 biology-14-00915-f013:**
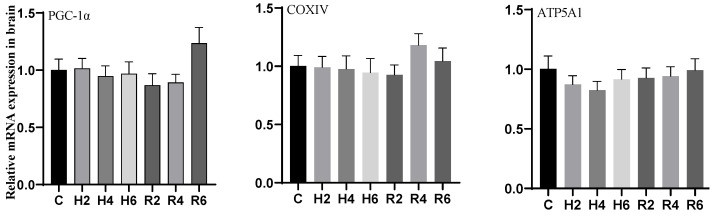
Effects of hypoxia (2 h, 4 h, 6 h) and reoxygenation (2 h, 4 h, 6 h) on brain mitochondrial damage-related genes of hybrid catfish.

**Figure 14 biology-14-00915-f014:**
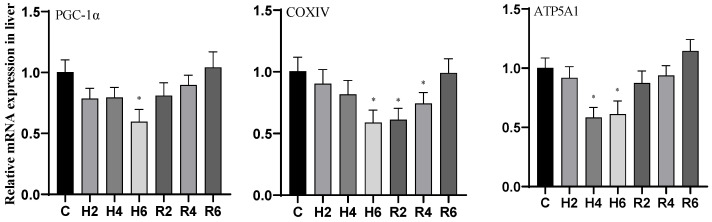
Effects of hypoxia (2 h, 4 h, 6 h) and reoxygenation (2 h, 4 h, 6 h) on liver mitochondrial damage-related genes of hybrid catfish. Note: * at *p* < 0.05.

**Figure 15 biology-14-00915-f015:**
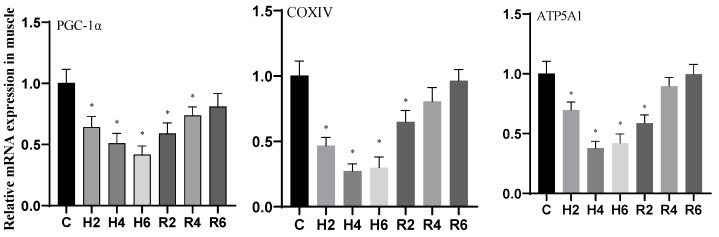
Effects of hypoxia (2 h, 4 h, 6 h) and reoxygenation (2 h, 4 h, 6 h) on muscle mitochondrial damage-related genes of hybrid catfish. Note: * at *p* < 0.05.

**Figure 16 biology-14-00915-f016:**
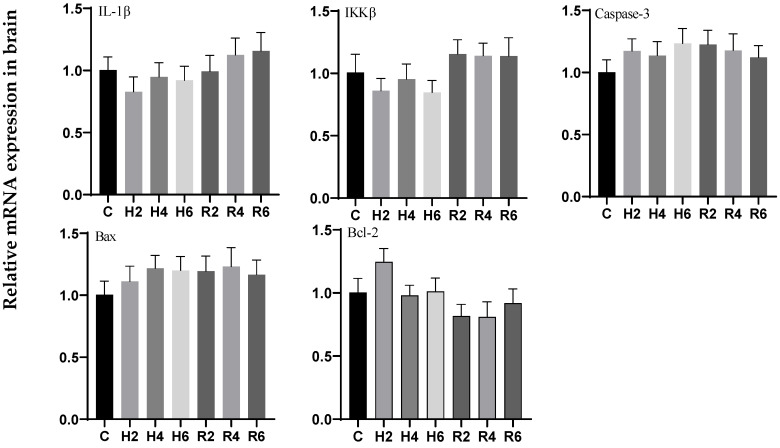
Effects of hypoxia (2 h, 4 h, 6 h) and reoxygenation (2 h, 4 h, 6 h) on apoptosis and inflammation-related genes in brain cells of hybrid catfish.

**Figure 17 biology-14-00915-f017:**
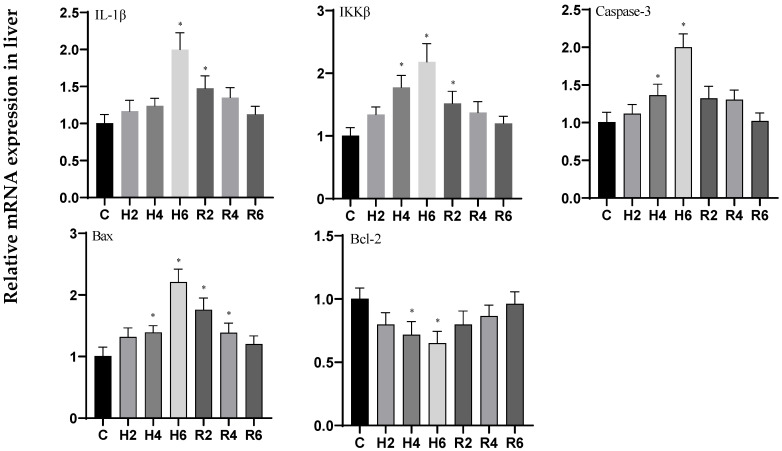
Effects of hypoxia (2 h, 4 h, 6 h) and reoxygenation (2 h, 4 h, 6 h) on apoptosis and inflammation-related genes in liver cells of hybrid catfish. Note: * at *p* < 0.05.

**Figure 18 biology-14-00915-f018:**
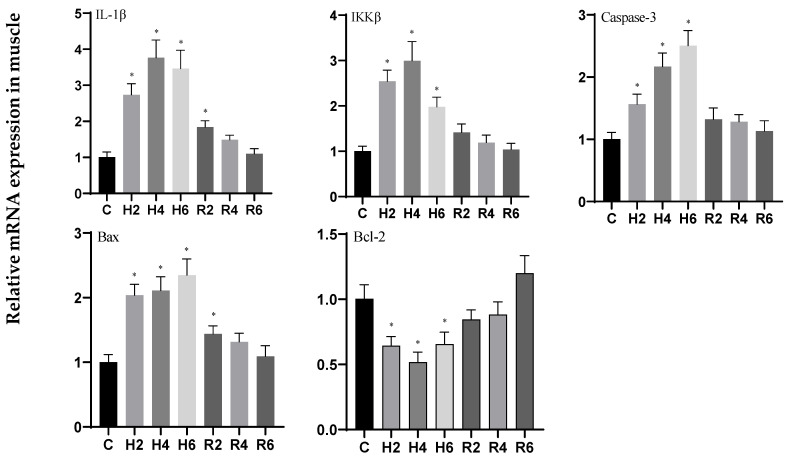
Effects of hypoxia (2 h, 4 h, 6 h) and reoxygenation (2 h, 4 h, 6 h) on apoptosis and inflammation-related genes in muscle cells of hybrid catfish. Note: * at *p* < 0.05.

**Table 1 biology-14-00915-t001:** Description of the primers used in this study.

Gene	Primer Sequence (5′-3′)	Product Size (bp)	Accession No. Or Publication	Amplification Efficiency
*HIF-1α*	Forward: CTGGAAAGAGGGCTAAGGTG	151	XP_027023235.1	98.26
	Reverse: AGTGACGGTCCTGAATAGGG			
*PFKL*	Forward: TTGTGGATACCTGGCAACCA	188	XP_027018682.1	98.81
	Reverse: AGTCGGTGCTGTATTGTGGA			
*PK*	Forward: GGCTAATGCTGTTCTGGATG	166	XP_027029118.1	97.74
	Reverse: GGGTCGGTAGAGTAGGCTGT			
*HK1*	Forward: CGAGACTGTGGACGGAGACG	141	XP_027030735.1	98.12
	Reverse: TTTGCCAGGGTTGAGGGAGA			
*CS*	Forward: GATGGGTGAAGTGGGGAAGA	154	XP_027016864.1	98.63
	Reverse: GGTGTTGAGCAGGAAGTTGG			
*LDHA*	Forward: ATCTGGACTCTGCTCGGTTC	112	XP_027028964.1	98.73
	Reverse: CATACAGGCACGCTTGAGTC			
*SOD1*	Forward: GTAATGTGACTGCCGGTTCC	106	AOQ25512.1	97.72
	Reverse: TGAATCACCATGGTCCTCCC			
*SOD2*	Forward: CCAAAGGTGACGTGACAACA	153	AOQ25511.1	97.95
	Reverse: AATCACGCTTAATGGCCTCC			
*GSH-PX*	Forward: GAATGGGAAAGACGCTCACC	118	XP_026771571.1	98.25
	Reverse: GCACACAGGACTCCAGATGA			
*CAT*	Forward: TCCCACACCTTCAAGCTGAT	156	XP_027019602.1	98.17
	Reverse: GGAGTTGTACAGGTCACGGA			
*Bcl-2*	Forward: CTTGTACCGACCCGACTTTG	122	XM_027142465.2	97.88
	Reverse: GTCCCCAGTTGATCCCGTC			
*Bax*	Forward: TGGAGATGAACTGGACAGCA	128	XP_027009035.1	97.52
	Reverse: TGCCCCAGTTAAACTTCCCA			
*IL-1β*	Forward: TTAGGCATAGAGGAGGTAAAAGAC	123	XP_027008231.1	97.92
	Reverse: TTCACCGACTCGAAGGTGTT			
*IKKβ*	Forward: GCGAGAGATGGAGCAAACTG	183	XM_047823116.1	98.59
	Reverse: TTCTCTCTCAGCTTGCGGAA			
*Caspase 3*	Forward: AGACCTGGACCCTGGTATTG	138	XP_026990614.1	97.64
	GAGTAGTAACCTGGTGCTGTAGAA			
*PGC-1α*	Forward: TTGACCACCACAGGCATAGT	112	XM_027150337.2	98.87
	Reverse: GTCCTCTCTTCTGGTGGCAT			
*COXIV*	Forward: ACTGAATCCGGAGCAGAAGT	165	XM_027177342.2	98.45
	Reverse: AACAACATTCCAGCGACCAC			
*ATP5A1*	Forward: ACTGATACCGGTGCCATGAA	137	XM_027133276.2	97.29
	Reverse: ATAAGTGCGTGTTTGCCGTT			
*β-actin*	Forward: CACTGCTGCCTCTTCCTC	181	XM_027148463.2	98.62
	Reverse: ATCCACATCGCACTTCAT			

## Data Availability

The data that support the findings of this study are available in [Mendeley Data] at [https://data.mendeley.com/preview/wn9fp3xfbm?a=597e50bd-e39e-47a3-bb16-ad66f914bda1], accessed on 23 May 2025.

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
