# Peer review of "Effects of Hypoxia and Reoxygenation on Hypoxia-Responsive Genes, Physiological and Biochemical Indices in Hybrid Catfish (Pelteobagrus vachelli ♀ × Leiocassis longirostris ♂)"

_biology, 2025, doi:10.3390/biology14080915_

Round 1

Reviewer 1 Report (Previous Reviewer 2)

Comments and Suggestions for Authors

The manuscript titled 'Effects of Hypoxia and Reoxygenation on Hypoxia-Responsive Genes, Physiological and Biochemical Indices in Hybrid Catfish (Pelteobagrus vachelli♀ × Leiocassis longirostris♂)' has addressed the revisions point by point, presents valuable data, and is now generally well-structured. I therefore agree to recommend its publication in this journal.

Author Response

Thank you to the reviewers for their recognition of the article. Thank you to the reviewers for their professionalism and conscientiousness.

Reviewer 2 Report (Previous Reviewer 4)

Comments and Suggestions for Authors

A brief summary:

This MS by Yan et al. was undertaken to investigate the effects of hypoxic stress (0.7 mg/L) and reoxygenation (7.0 mg/L) on oxygen-sensing proteins, respiratory metabolism, oxidative stress, mitochondrial function, inflammatory response, and cell apoptosis in the brain, liver, and tissues of the hybrid catfish (Pelteobagrus vachelli♀ × Leiocassis longirostris♂). These findings help explain how hybrid catfish tolerate fluctuating oxygen levels, which is crucial for improving aquaculture practices and ensuring fish health in changing environments.

Overall, I thought this was a well-executed study in a system with limited previous knowledge of this level on this specific topic. I appreciated their multi-faceted approach (Biochemistry and Molecular biology) and the time course involved in this work and think it add significant merit to their work. I think the Materials and Methods part was clea. I also think that their overall conclusions were related to results. Therefore, I think this is good work, but should undergo minor revision before acceptance.

  • General concept comments.

I think the author should seek help with writing in English, I hope round of editing with English native language would further improve the quality of the writing. The English grammar should be checked again. I also have a couple suggestions.

  • Specific comments

Line 68-99 PFKL, HK1, PK, LDHA, CS, the first appearance should use the full name.

Line 108 The author should introduce the parental fish of hybrid catfish. How to hybridize? How old is the parental fish? Which one is paternal and which one is maternal?

Line 137 “dissolved oxygen (DO)” change to “DO”

Line 129 “Tissues”? this is too simple. Which tissues (brain, liver…) and how to sample? make it clear.

Line 142 “Total RNA was isolated” the way introduced here was too easy, make it clear for repeat experiments.

Line 162 “cDNA synthesis” the way introduced here was too easy, make it clear for repeat experiments.

Author Response

Comments 1: Line 68-99 PFKL, HK1, PK, LDHA, CS, the first appearance should use the full name.

Response 1: Thank you to the reviewers for your comments. We have added the full names of these genes. (Line68-99)

Comments 2: Line 108 The author should introduce the parental fish of hybrid catfish. How to hybridize? How old is the parental fish? Which one is paternal and which one is maternal?

Response 2: We sincerely thank you for your valuable suggestions. We have made a clear description in the corresponding part of the text. (Line97-98)

Comments 3: Line 137 “dissolved oxygen (DO)” change to “DO”

Response 3: We are very grateful for the reviewers' comments, We have modified it. (Line139)

Comments 4: Line 129 “Tissues”? this is too simple. Which tissues (brain, liver…) and how to sample? make it clear.

Response 4: Thanks to the reminder from the reviewers, we have already provided a detailed description in the article.(Line138-159)

Comments 5: Line 142 “Total RNA was isolated” the way introduced here was too easy, make it clear for repeat experiments.

Response 5: Thank you for the reviewers' comments. We have reelaborated on this part. (Line161-172)

Comments 6: Line 162 “cDNA synthesis” the way introduced here was too easy, make it clear for repeat experiments.

Response 6: Thank you for the reviewers' comments. We have reelaborated on this part. (Line161-172)

Comments 7: I think the author should seek help with writing in English, I hope round of editing with English native language would further improve the quality of the writing. The English grammar should be checked again. I also have a couple suggestions.

Response 7: Thank you to the reviewers for your comments. We have corrected the grammar using grammariy software and invited friends majoring in foreign languages to help revise our article. We hope the revised manuscript will be accepted by you.

We have made every effort to improve the manuscript and have incorporated revisions marked with yellow highlights in the revised paper. These modifications do not affect the core content or overall structure of the work. We sincerely appreciate the diligent efforts of the editors and reviewers and hope that the revisions meet their approval. Once again, we extend our gratitude for your valuable comments and suggestions.

This manuscript is a resubmission of an earlier submission. The following is a list of the peer review reports and author responses from that submission.

Round 1

Reviewer 1 Report

Comments and Suggestions for Authors

The present study explored how hybrid catfish respond to low oxygen levels (hypoxia) and subsequent recovery (reoxygenation), a common challenge in aquatic environments due to factors like pollution and climate change. We examined changes in genes and physiological processes in the brain, liver, and muscle tissues of the fish under controlled hypoxia and reoxygenation conditions. We found that low oxygen triggered adaptive responses, such as switching energy production from aerobic to anaerobic pathways in the liver, while the brain reduced anaerobic activity to avoid harmful lactate buildup. Muscle tissue showed the most severe effects, including oxidative stress, mitochondrial damage, and increased cell death. After oxygen levels were restored, the fish recovered, but signs of stress lingered in some tissues. The research work is systematic and meticulous. However, there are still some problems that need to be corrected in the article.

Major comment

Line28-29

The organization name is unclear due to repetition. The phrase 'brain, liver, and tissues' should specify the exact types of tissues, and it is recommended to change it to 'brain, liver, and muscle tissues'.

Line41-42

There is a lack of data to support the description of brain tissue metabolism. It is necessary to provide specific indicators for 'weakening of anaerobic glycolysis' (such as changes in LDH activity).

Line 52: Redundant keyword repetition. Suggest deleting 'oxygen-sensing protein' and retaining 'hypoxia-responsive genes'. Add 'metabolic reprogramming'.

Line62-64

The research background needs to be supplemented with the latest literature. It is recommended to include references to recent studies (within the last three years) on the molecular mechanisms of fish adaptation to hypoxia, such as (DOI: 10.3390/genes15080987)

Line104

Sample size description is incomplete; it is necessary to include an explanation of the number of biological replicates: '3 biological replicates (n=5 fish per replicate)'

Line115-121

The description of the preliminary experiment method is insufficient. It should be supplemented with parameters for controlling the nitrogen flow rate and methods for maintaining oxygen stability.

Line136-138

The description of reoxygenation treatment is vague. Specific details on the reoxygenation rate (such as the increase in dissolved oxygen per hour) and the frequency of water quality monitoring are required.

Line154-163

The enzyme activity assay method is too brief. It is recommended to provide additional parameters for homogenization and centrifugation (such as g-force values) as well as information on the spectrophotometer model.

Line174-204

The results section lacks statistical indicators. Significance markers (e.g., *P<0.05, **P<0.01) need to be added to the chart descriptions.

Line175-185

The gene expression trend does not match the chart. Verify if the trend of HK1 gene expression from H2 to R4 in Figure 1 aligns with the textual description.

Line196-204

Muscle tissue analysis results are weak. It is recommended to supplement the correlation analysis between mitochondrial function genes such as ATP5A1 and metabolic conversion.

Line428

The discussion section lacks cross-species comparisons. It is necessary to add comparisons of low-oxygen response mechanisms with model fish species such as zebrafish and tilapia.

On line 137, '7.0mg/L' should be modified to '7.0 mg/L' with a space added.

The term 'GSH-PX' first appearing at line 34 has not been provided with its full name.

The ethical approval number for experimental animals has not been mentioned and needs to be provided.

Table 1 does not list primer efficiency validation data; amplification efficiency and melting curve analysis results need to be supplemented.

Reviewer 2 Report

Comments and Suggestions for Authors

Comments and Suggestions for Authors (Major Revision)

The manuscript presents data from a “Effects of Hypoxia and Reoxygenation on Hypoxia-Responsive 2 Genes, Physiological and Biochemical Indices in hybrid catfish 3 (Pelteobagrus vachelli♀×Leiocassis longirostris♂)”. The main flow of the manuscript is the Results section, where the authors are not restricted to significant findings, but also use not significant effects as important results. Unfortunately, this also has an effect on Discussion section, where conclusions are not totally supported by the results.  Finally, the use of English language should be revised to correct many errors in expression and syntax (not all mentioned below in specific comments).

Specific comments

  1. The authors should follow the guidelines of the journal. It is specified that the abstract should have a max of 200 words, this one have 291.
  2. add a small background to the abstract at the beginning.
  3. Summarize the simple summary
  4. Please updated the reference no: 9,33, 34, 36, 18,21, 22.
  5. As you research include with live animals. So, provide Ethical Statement.
  6. Please if possible provide growth parameters. Becaue mybe the also effect the growth (: IBW, Initial average body weight; FBW, Final average body weight; WGR, weight gain rate; SGR, specific growth rate; MFI, mean feed intake; FE, feed efficiency; CF, condition factor; HSI, hepatosomatic index; IPR, intraperitoneal fat ratio; VSI, viscerosomatic index rate; SR, survival rate).
  7. In line 88,89 Please write italic the scientific name.
  8. Some problem in line 90, 91 and so on. Please confirm it all manuscript.
  9. In Line 107-109 Please add density 100 fish m3
  10. Please provide formula in tabulated form.
  11. Explain tanks number, number of fish per tanks and
  12. Explain replication
  13. Provide Product Size (bp) of gene. Follow this research article (https://doi.org/10.3390/ani15060810)
  14. Provide Accession No Or Publication of gene. Follow this research article (https://doi.org/10.3390/ani15060810)
  15. Provide Amplification Efficiency of gene. Follow this research article (https://doi.org/10.3390/ani15060810)
  16. Improve result section accordingly journal requirements
  17. Line 433 reference for this paragraph.
  18. Line 505 provide reference for this paragraph.
  19. Line 506 provide reference for this paragraph.
  20. Line 507-513 the study is related with fish but the reference is related to rate. Please try to provide fish related or even aquatic animals related references.
  21. Line 518-520 provide reference for this paragraph. Follow this research article (https://doi.org/10.3390/ani15060810)

Line 545-559 The Conclusion section is quite lengthy. Please condense and refine it.

Comments on the Quality of English Language

English needs to improve.

Reviewer 3 Report

Comments and Suggestions for Authors

This paper aims to characterize the effect of hypoxia and subsequent re-oxygenation on a hybrid variety of catfish (Pelteobagrus vachelliâ™€× Leiocassis longirostris ♂). The authors summarize that understanding the changes in pathophysiology in different tissues of fish during hypoxia will help in fish breeding and food security. They looked at RNA levels and changes in enzymatic activities of oxygen sensing gene, respiratory metabolism, oxidative stress, apoptosis and other pathway related genes during hypoxia and reoxygenation conditions. They observed that in liver, anerobic metabolism is activated while in brain aerobic glycolysis is subdued. They also noted that oxidative stress significantly increases.

Simple summary and abstract are adequate.

Introduction could benefit from a better explanation of why all the genes in the pathways are chosen. Will also help unversed readers to have a schematic of pathways and why specific genes were selected.

Materials and Methods - This section is also adequate. Will be good to have a better description of the kit used for enzymatic activity levels evaluation

Results section is well described and clearly written. Data is plotted well and sectioned into appropriate sub categories

Discussion and conclusion read well. Impact on all the genes and subsequently pathways is summarized well.

Reviewer 4 Report

Comments and Suggestions for Authors

A brief summary:

This MS by Yan et al. was undertaken to investigate the effects of hypoxic stress (0.7 mg/L) and reoxygenation (7.0 mg/L) on oxygen-sensing proteins, respiratory metabolism, oxidative stress, mitochondrial function, inflammatory response, and cell apoptosis in the brain, liver, and tissues of the hybrid catfish (Pelteobagrus vachelli♀ × Leiocassis longirostris♂). The results demonstrate that hypoxia significantly activates HIF-1α expression and induces notable shifts in energy metabolism pathways. The liver exhibits a transition from aerobic to anaerobic metabolism under hypoxia, whereas the brain shows reduced anaerobic glycolysis in later stages, and the muscle experiences coordinated suppression of glycolysis and aerobic metabolism. Additionally, muscle tissue exhibits more pronounced oxidative stress, mitochondrial dysfunction, inflammatory cascades, and apoptosis during hypoxia/reoxygenation. These findings reveal the multi-tissue response mechanisms of Hybrid catfish under hypoxia and reoxygenation.

Overall, I thought this was a well-executed study in a system with limited previous knowledge of this level on this specific topic. I appreciated their multi-faceted approach (Biochemistry and Molecular biology) and the time course involved in this work and think it add significant merit to their work. I think the Materials and Methods part was not clear, and should supplement some necessary information such as sampling number and exacted tissues…. I also think that their overall conclusions were related to results. Therefore, I think this is good work, but should undergo major revision before acceptance.

  • General concept comments.

I think the author should seek help with writing in English, I hope round of editing with English native language would further improve the quality of the writing. I find some obvious grammatical errors. The English grammar should be checked again.

I also have a couple suggestions.

  • Specific comments

Line 88-89 The fish name should be italic in Latin.

Line 105 The author should introduce the parental fish of hybrid catfish. How to hybridize? How old is the parental fish and hybrid catfish? as well as other information.

Line 110 Did the dissolved oxygen (DO) concentration not change and at 7.0 mg/L all the time? Is inflation used in the recirculating tank? What about the pH change?

Line 112 “formulated artificial feed” make it clear, including type, model, and source company.

Line 116 “the aeration and water circulation systems” However, there is no aeration information in Experimental Materials.

Line 119 “LDO101 probe” make it clear, including type, model, and source company.

Line 125-127 The author should show the fish number, size (TL and BW) in each group. Because we only know the total fish was 150 and body weight of (20 ± 1.1) g and an average body length of (12 ± 0.7). (Line 107),

Line 129 “Tissues”? this is too simple. Which tissues (brain, liver…) and how to sample? make it clear.

Line 135-136 How many fish were sampled here at 2 h, 4 h, and 6 h each time?

Line 138 How many fish were sampled here at 2 h, 4 h, and 6 h each time?

Line 141 “Samples” what samples?

Line 142 “a rapid RNA extraction kit” make it clear, including model and source company

Line 152 “2-ΔΔCt” change to “2-ΔΔCt

Line 160-164 There are many Enzyme here, but we do not know the role and function. I suggest the author simply introduce the role and function of most Enzymes in Part 1. Introduction.

Line 176 “P < 0.05.” P italic. Other similar ones need to be modified.

The Fig 2 can't be seen clearly.

Comments on the Quality of English Language

I think the author should seek help with writing in English, I hope round of editing with English native language would further improve the quality of the writing. I find some obvious grammatical errors. The English grammar should be checked again.

Round 2

Reviewer 1 Report

Comments and Suggestions for Authors

No comment

Author Response

Thank you very much.

Reviewer 2 Report

Comments and Suggestions for Authors

Comments and Suggestions for Authors (Major Revision)

The manuscript presents data from a “Effects of Hypoxia and Reoxygenation on Hypoxia-Responsive 2 Genes, Physiological and Biochemical Indices in hybrid catfish 3 (Pelteobagrus vachelli♀×Leiocassis longirostris♂)”. The main flow of the manuscript is the Results section, where the authors are not restricted to significant findings, but also use not significant effects as important results. Unfortunately, this also has an effect on Discussion section, where conclusions are not totally supported by the results.  Finally, the use of English language should be revised to correct many errors in expression and syntax (not all mentioned below in specific comments).

Specific comments

  1. The authors should follow the guidelines of the journal. It is specified that the abstract should have a max of 200 words, this one have 291.
  2. add a small background to the abstract at the beginning.
  3. Summarize the simple summary
  4. Please updated the reference no: 9,33, 34, 36, 18,21, 22.
  5. As you research include with live animals. So, provide Ethical Statement.
  6. Please if possible provide growth parameters. Becaue mybe the also effect the growth (: IBW, Initial average body weight; FBW, Final average body weight; WGR, weight gain rate; SGR, specific growth rate; MFI, mean feed intake; FE, feed efficiency; CF, condition factor; HSI, hepatosomatic index; IPR, intraperitoneal fat ratio; VSI, viscerosomatic index rate; SR, survival rate).
  7. In line 88,89 Please write italic the scientific name.
  8. Some problem in line 90, 91 and so on. Please confirm it all manuscript.
  9. In Line 107-109 Please add density 100 fish m3
  10. Please provide formula in tabulated form.
  11. Explain tanks number, number of fish per tanks and
  12. Explain replication
  13. Provide Product Size (bp) of gene. Follow this research article (https://doi.org/10.3390/ani15060810)
  14. Provide Accession No Or Publication of gene. Follow this research article (https://doi.org/10.3390/ani15060810)
  15. Provide Amplification Efficiency of gene. Follow this research article (https://doi.org/10.3390/ani15060810)
  16. Improve result section accordingly journal requirements
  17. Line 433 reference for this paragraph.
  18. Line 505 provide reference for this paragraph.
  19. Line 506 provide reference for this paragraph.
  20. Line 507-513 the study is related with fish but the reference is related to rate. Please try to provide fish related or even aquatic animals related references.
  21. Line 518-520 provide reference for this paragraph. Follow this research article (https://doi.org/10.3390/ani15060810)
  22. Line 545-559 The Conclusion section is quite lengthy. Please condense and refine it.

Author Response

Comments 1: The authors should follow the guidelines of the journal. It is specified that the abstract should have a max of 200 words, this one have 291.

Response 1: We have rewritten this part based on the reviewers' comments. (Line26)

Comments 2: add a small background to the abstract at the beginning.

Response 2: We added a small background at the beginning of the abstract based on the reviewers' opinions. (Line26)

Comments 3: Summarize the simple summary.

Response 3: We have rewritten this part based on the reviewers' comments. (Line26)

Comments 4: Please updated the reference no: 9,33, 34, 36, 18,21, 22.

Response 4: We updated some reference numbers based on the reviewers' comments. (Line75,490)

Comments 5: As you research include with live animals. So, provide Ethical Statement.

Response 5: This study and the included experimental procedures have been approved by the Animal Ethics Committee of Nanjing Normal University(approval no. SYXK [Jiangsu] 2015-0028). (Line587-588)

Comments 6: Please if possible provide growth parameters. Becaue mybe the also effect the growth (: IBW, Initial average body weight; FBW, Final average body weight; WGR, weight gain rate; SGR, specific growth rate; MFI, mean feed intake; FE, feed efficiency; CF, condition factor; HSI, hepatosomatic index; IPR, intraperitoneal fat ratio; VSI, viscerosomatic index rate; SR, survival rate).

Response 6: We agree with the reviewer that more data would be useful to understand. However, limited by the laboratory conditions, it is impractical to implement the related experiments. In the future, we would pay more attention to improve the experimental prototype.

Comments 7: In line 88,89 Please write italic the scientific name.

Response 7: We were really sorry for our careless mistakes.Thankyou for your reminder. (Line90)

Comments 8: Some problem in line 90, 91 and so on. Please confirm it all manuscript.

Response 8: We were really sorry for our careless mistakes.Thankyou for your reminder. (Line91-96)

Comments 9: In Line 107-109 Please add density 100 fish m3.

Response 9: Thank you for the reminder from the reviewers. We have already added this content to the manuscript. (Line112-114)

Comments 10: Please provide formula in tabulated form.

Response 10: Thank you for the reminder from the reviewers. We have already added this content to the manuscript. (Line112-114)

Comments 11: Explain tanks number, number of fish per tanks and explain replication.

Response 11: We think this is an excellent suggestion. We have added the number of fish tanks and the number of fish in each tank. (Line111-114)

Comments 12: Provide Product Size (bp) of gene. Follow this research article (https://doi.org/10.3390/ani15060810).

Response 12: We are very grateful for the reviewers' comments. We have supplemented the relevant content in Table 1. (Line172)

Comments 13: Provide Accession No Or Publication of gene. Follow this research article (https://doi.org/10.3390/ani15060810).

Response 13: We are very grateful for the reviewers' comments. We have supplemented the relevant content in Table 1. (Line172)

Comments 14: Provide Amplification Efficiency of gene. Follow this research article (https://doi.org/10.3390/ani15060810).

Response 14: We are very grateful for the reviewers' comments. We have supplemented the relevant content in Table 1. (Line172)

Comments 15: Improve result section accordingly journal requirements.

Response 15: The authors wholeheartedly concur with the reviewer's astute observation that additional studies, data, or further experimental results would indeed add substantial value to the manuscript. However, limited by the laboratory conditions, it is impractical to implement the related experiments. In the future, we would pay more attention to improve the experimental prototype.

Comments 16: Line 433 reference for this paragraph.

Response 16: We sincerely thank you for your valuable suggestions. We added the corresponding references in the manuscript. (Line454)

Comments 17: Line 505 provide reference for this paragraph.

Response 17: We sincerely thank you for your valuable suggestions. We added the corresponding references in the manuscript. (Line526)

Comments 18: Line 506 provide reference for this paragraph.

Response 18: We sincerely thank you for your valuable suggestions. We added the corresponding references in the manuscript. (Line528)

Comments 19: Line 507-513 the study is related with fish but the reference is related to rate. Please try to provide fish related or even aquatic animals related references.

Response 19: We sincerely thank you for your valuable suggestions. However, there are relatively few studies related to aquatic animals here, and no more references can be provided. (Line531)

Comments 20: Line 518-520 provide reference for this paragraph. Follow this research article (https://doi.org/10.3390/ani15060810).

Response 20: We sincerely thank you for your valuable suggestions. We added the corresponding references in the manuscript. (Line541)

Comments 21: Line 545-559 The Conclusion section is quite lengthy. Please condense and refine it.

Response 21: We have condensed the conclusion part based on the suggestions of the reviewers. (Line567)

We have made every effort to improve the manuscript and have incorporated revisions marked with yellow highlights in the revised paper. These modifications do not affect the core content or overall structure of the work. We sincerely appreciate the diligent efforts of the editors and reviewers and hope that the revisions meet their approval. Once again, we extend our gratitude for your valuable comments and suggestions.

Reviewer 4 Report

Comments and Suggestions for Authors

The author has made point-to-point revisions according to the reviewer's suggestions, and I agree to publish after language polishing.

Author Response

Thank you very much.